# Disentangling the consequences of type 2 diabetes on targeted metabolite profiles using causal inference and interaction QTL analyses

**Ozvan Bocher**[1]*, **Archit Singh**[1,2,3], **Yue Huang**[1,3], **Urmo Võsa**[4], **Ene Reimann**[1,4], **Ana Arruda**[1,2,3], **Andrei Barysenska**[1], **Anastassia Kolde**[4,5], **Nigel W. Rayner**[1], **Estonian Biobank research team**[¶], **Tõnu Esko**[4], **Reedik Mägi**[4], **Eleftheria Zeggini**[1,6]

1 Institute of Translational Genomics, Helmholtz Zentrum München- German Research Center for Environmental Health, Neuherberg, Germany, 2 Munich School for Data Science (MUDS), Helmholtz Zentrum München- German Research Center for Environmental Health, Neuherberg, Germany, 3 Technical University of Munich (TUM), TUM School of Medicine and Health, Graduate School of Experimental Medicine, Munich, Germany, 4 Estonian Genome Centre, Institute of Genomics, University of Tartu, Tartu, Estonia, 5 Institute of Mathematics and Statistics, University of Tartu, Tartu, Estonia, 6 TUM school of medicine and health, Technical University Munich and Klinikum Rechts der Isar, Munich, Germany

¶ Membership of Estonian Biobank research team is provided in the Acknowledgements
* ozvan.bocher@helmholtz-munich.de

## Abstract

Circulating metabolite levels have been associated with type 2 diabetes (T2D), but the extent to which T2D affects metabolite levels and their genetic regulation remains to be elucidated. In this study, we investigate the interplay between genetics, metabolomics, and T2D risk in the UK Biobank dataset using the Nightingale panel composed of 249 metabolites, 92% of which correspond to lipids (HDL, IDL, LDL, VLDL) and lipoproteins. By integrating these data with large-scale T2D GWAS from the DIAMANTE meta-analysis through Mendelian randomization analyses, we find 79 metabolites with a causal association to T2D, all spanning lipid-related classes except for Glucose and Tyrosine. Twice as many metabolites are causally affected by T2D liability, spanning almost all tested classes, including branched-chain amino acids. Secondly, using an interaction quantitative trait locus (QTL) analysis, we describe four metabolites consistently replicated in an independent dataset from the Estonian Biobank, for which genetic loci in two different genomic regions show attenuated regulation in T2D cases compared to controls. The significant variants from the interaction QTL analysis are significant QTLs for the corresponding metabolites in the general population but are not associated with T2D risk, pointing towards consequences of T2D on the genetic regulation of metabolite levels. Finally, through differential level analyses, we find 165 metabolites associated with microvascular, macrovascular, or both types of T2D complications, with only a few discriminating between complication classes. Of the 165 metabolites, 40 are not causally linked to T2D in either direction, suggesting biological mechanisms specific to the occurrence of complications. Overall, this work provides a map of the consequences of T2D on Nightingale targeted metabolite levels and on their genetic regulation, enabling a better understanding of the T2D trajectory leading to complications.

**Data Availability Statement:** Summary statistics of the mQTL analysis in the second data release of the UKBB are available in the GWAS catalog and on the following FTP: https://ftp.ebi.ac.uk/pub/databases/gwas/summary_statistics/GCST90447001-GCST90448000/ and the GCP ID is GCP000941.

**Funding:** OB, AS, YH, ER and RM have received funding from the European Union's Horizon 2020 research and innovation programme under Grant Agreement No 101017802 (OPTOMICS). TE and UV are supported by Estonian Research Council grant PUT (PRG1291) and RM by Estonian Research Council grant PUT (PRG1911) and by Estonian Ministry of Education and Research Funding (TK214). The funders had no role in study design, data collection and analysis, decision to publish, or preparation of the manuscript.

**Competing interests:** The authors have declared that no competing interests exist.

## Author summary

Type 2 diabetes (T2D) is a complex disease that affects millions of people worldwide, with its progression to complications posing a significant health burden. While the genetics of the disease has been increasingly described through large efforts, the biological mechanisms behind these associations remain to be fully elucidated. Additionally, the consequences of the disease on molecular profiles, and how these changes could lead to further complications, remain poorly understood. Here, we have investigated the relationships between T2D, its complications and 249 targeted metabolites mostly focusing on lipids. We show a causal impact of 79 metabolites on T2D risk, while twice as many metabolites were causally affected by T2D liability. Further, we highlight four metabolites whose genetic regulation appears to be altered in T2D cases, which, along with additional lines of evidence, seems to be a consequence rather than a cause of T2D occurrence. Finally, we found that 165 metabolites were associated with the occurrence of T2D complications, with most also showing causal links to T2D, although 40 of them did not demonstrate such associations. Our study enhances our understanding of the metabolic pathways involved in the trajectory of T2D to complications and represents a first step towards the prevention of the development of complications in the future, a key public health priority.

## Introduction

Type 2 diabetes (T2D) is a common, complex disease, with a prevalence that is expected to increase dramatically, and for which the genetics has been largely described through genome-wide association study (GWAS) meta-analysis efforts [1–3]. The next step towards translating these associations into the clinic is to understand the biological mechanisms behind them. To this end, multi-omics data such as metabolite levels offer great promise, as they enable the study of molecular phenotypes closely implicated in the disease. A large number of metabolites have been associated with T2D, as highlighted in a recent study from Julkunen et al. [4], which described 230 metabolites as nominally associated with incident and prevalent T2D. Consistent associations between T2D risk and metabolites across studies mostly cover increases in various amino acid levels, especially branched-chain amino acids (BCAAs) [5–7], and dyslipidemia [4,8]. However, the predictive value of metabolite profiles in T2D risk is still debated. Improved prediction of various complex traits using metabolite profiles over genetic scores has been shown[9], but this improvement is rather limited when compared to classical risk factors including age, sex and family history of disease [10,11]. The limited prediction of metabolite profiles may be related to the uncertain causal role in modulating T2D risk, a question investigated using Mendelian randomization (MR) [12]. One particular example is BCAAs, for which some studies have described a causal effect of valine, leucine and/or isoleucine on T2D [13–15], while more recent evidence has shown no causal effect of BCAAs on T2D risk in the UK Biobank (UKBB) cohort [16]. Conversely, increasing evidence of a causal effect of T2D liability on metabolite profiles has been shown, with increased alanine levels caused by increased T2D liability being consistently reported [13,17,18]. Conflicting conclusions exist for other metabolites, and little is known about the role of genetic regulation in shaping these relationships. The genetic regulation of metabolite profiles is increasingly described, for example through the latest meta-analysis reporting over 400 independent loci associated with 233 metabolite levels [19]. These metabolite QTLs, also called mQTLs, are usually reported in the general population, and studies are starting to emerge on how they are modified by factors

such as diet [20]. However, no study to date has investigated the question of whether the genetic regulation of metabolite levels is affected by the occurrence of T2D.

A large part of the healthcare burden associated with T2D is due to subsequent complications of the disease. T2D complications span both microvascular complications, which refer to complications involving small vessels and have an estimated prevalence of 53%, and macrovascular complications which refer to complications involving large vessels such as arteries and veins with an estimated prevalence of 27% [21]. Increasing evidence is emerging that metabolite profiles are also associated with the risk of developing complications, such as the fatty acid biosynthesis pathway with retinal and renal complications [22], two of the main microvascular complications of T2D, n-3 fatty acids with T2D macrovascular complications [23], or amino acids with both types of T2D complications [24]. However, replication of these associations is still needed, as well as investigating whether the metabolites associated with T2D complications are distinct from the ones causally affected by T2D liability. Exploring the links between metabolite levels and T2D can help us gain insights into how these relationships may influence trajectories towards T2D complications, for which underlying biological mechanisms are still to be unraveled.

Here, we aim to address these questions and elucidate the metabolic consequences of T2D by investigating the links between metabolite profiles, genetics, and the risk of T2D and subsequent complications. For this, we considered profiles of 249 metabolites from the UKBB cohort [25] assessed through the Nightingale panel, mostly composed of lipids and lipoproteins classes (around 92% of metabolites), and covering additional classes such as amino acids (10 metabolites), ketone bodies (4 metabolites), glycolysis-related metabolites (4 metabolites), and markers of inflammation (1 metabolite). These levels were characterized for almost 275,000 participants through two releases, which were here considered for discovery (n = 117,967 individuals) and replication (n = 156,385 individuals), or meta-analyzed when statistically appropriate. We first performed a bi-directional two sample MR analysis, using non-overlapping datasets for the SNP-exposure and SNP-outcome associations to limit potential bias [12]. Previous MR studies performed in the UKBB cohort have used only the first release of data for the 249 metabolites and were either limited to a few metabolites, or only investigated a single causal direction [16,18,26]. Secondly, we performed an interaction mQTL analysis to assess whether there is a different genetic regulation of metabolite levels between T2D patients and controls. Finally, we investigated how metabolite profiles are associated with T2D complications, and whether these associated metabolites overlap with the metabolites causally affected by T2D liability.

## Methods

### Ethics statement

Individual level data analysis in the Estonian Biobank was carried out under ethical approval [nr 1.1-12/3337] from the Estonian Committee on Bioethics and Human Research (Estonian Ministry of Social Affairs), using data according to release application [nr 6-7/GI/15486 T17] from the Estonian Biobank.

### Data and quality control

Genetic data from genotyping arrays and imputation are available for over 500,000 participants in the UKBB cohort. Quality control (QC) was performed at the variant level and at the sample (S1 Appendix). We selected only variants with a minor allele frequency greater than 1% and with an imputation INFO score greater than 0.8. To minimize potential bias in MR due to differences between the exposure and outcome datasets which could be related to having different ancestry background in the two datasets, we chose to focus on unrelated

individuals of European ancestry. In total, 408,194 individuals and 9,572,578 variants remained after QC.

A total of 249 metabolite levels obtained from the Nightingale platform using nuclear magnetic resonance (NMR) spectroscopy is available in the UKBB cohort as part of two different releases covering a total of 274,352 individuals: 117,967 in the first release, and 156,385 in the second release. Absolute concentrations cover 168 metabolites, with an additional set of 81 metabolite ratios and percentages, which are further derived from the absolute concentrations (https://biobank.ndph.ox.ac.uk/showcase/refer.cgi?id=1682). To perform metabolite QC, we used the ukbnmr R package (version 1.5) specifically developed to remove technical variations from NMR metabolite measurements in the UKBB [27]. For each release, we considered only metabolite data at the first timepoint, T0, corresponding to a total of 227,607 individuals and 249 metabolites after applying the ukbnmr QC (S1 Appendix). All analyses have been performed on inverse normal transformed values to obtain normally distributed metabolite levels [28], which correspond to 'metabolite levels' for the rest of the manuscript.

## Definition of phenotypes

**T2D status.** T2D status was defined based on the UKBB field 130708, which corresponds to the first date of T2D report (self-reported or ICD10 code). Status was defined at T0 to analyze metabolite profiles in the light of T2D status at the time of profiling (Fig A in S2 Appendix), prevalent cases corresponding to individuals having a T2D diagnosis date before the date of blood sample collection on which metabolites were assayed, and incident cases to individuals having a T2D diagnosis date after the date of blood sample collection. Considering that T2D is often diagnosed with a few years delay [29], we considered HbA1c levels in addition of the field 130708 to recover individuals likely having undiagnosed T2D at T0 among T2D incident cases. We removed from the analysis individuals with any mention of T1D (ICD10 code E10*, field 130706) or gestational diabetes (ICD10 code O24*, field 132202), and individuals diagnosed with T2D before the age of 36 years, in accordance with previous guidelines [30]. Individuals with a mention of T1D or gestational diabetes were also removed from the controls. The final number of T2D cases and controls included in the analyses, passing the genetic data QC and having measured metabolite levels are 3,088 and 88,244 for the first release and 4,302 and 118,555 for the second release, respectively.

**T2D complications.** We categorized prevalent cases of T2D at T0 into four mutually exclusive complication groups based on the occurrence of generalized vascular complication events: "microvascular", "macrovascular", "micro and macrovascular" complications, and "no complications". "Microvascular" and "macrovascular" complications were defined based on ICD10 codes (Table A in S3 Appendix). Individuals with an ICD10 code for both types of complications were attributed to the "micro and macrovascular" group and removed from the two other complication groups. We selected individuals with a diagnosis date of complications before the date of metabolite sample collection but after the T2D diagnosis date. If individuals had multiple ICD10 codes for the "microvascular" or the "macrovascular" complication group, the date of the first event was considered. For the "micro and macrovascular" group, the latest date between the "microvascular" and the "macrovascular" onset was considered. The total number of individuals in each complication group assayed in each of the release datasets is presented in Table 1.

**Table 1. Number of T2D patients per complication group for each of the released dataset.**

| Metabolite data release | No complications | Microvascular | Macrovascular | Micro- and macrovascular |
|---|---|---|---|---|
| 1st release dataset | 1268 | 83 | 243 | 57 |
| 2nd release dataset | 1701 | 99 | 356 | 76 |

## Statistical analyses

**General considerations.** All the analyses have been performed using R version 4.2.0 and adjusted for age, sex, and fasting time (only available in the UKBB), as well as for additional covariates in the analyses involving genetic data (Table B in S3 Appendix). To correct our analyses for multiple testing, we considered FDR-adjusted p-values (Benjamini-Hochberg method, referred to as q-values) to assess significance at 5%. All analyses were performed on the two release datasets separately and meta-analyzed, except for the MR analyses where the same T2D summary statistics were used for both MR analyses on the two release sets (details below).

**Mendelian randomization.** Two sample bi-directional MR was performed to assess the causal effects of metabolite levels on T2D risk (forward MR) and the causal effects of T2D liability on metabolite levels (reverse MR). MR was run separately on the two released datasets as it was not possible to perform meta-analysis due to the same outcome data being used in the two MR analyses, namely T2D DIAMANTE meta-analysis. For the first release dataset, we used mQTL summary statistics from Borges et al.[31], which correspond to the QTL map of the 249 metabolites from the first data release that has been generated in the UK Biobank. For the second release, we performed a mQTL analysis using the REGENIE software with the--qt option considering the INT metabolite levels as the phenotype (version 2.2.4) [32]. For the associations between genetic variants and T2D, we used the DIAMANTE 2018 meta-analysis [2], restricted to European ancestry samples, and without the UKBB cohort to avoid sample overlap between the exposure and the outcome data.

Instrumental variables (IVs) were selected from independent variants defined through LD-based clumping using plink [33] with the following parameters: $R^2 < 0.001$ in windows of 10Mb, a p-value threshold of $2.54 \times 10^{-10}$ for the metabolite levels and of $5 \times 10^{-8}$ for T2D. The threshold of $2.54 \times 10^{-10}$ is a genome-wide threshold corrected by the number of effective tests [34], estimated at 197, which enables to correct the analyses for multiple testing while considering the correlation between the metabolites. For each of these independent variants, the strength of association with the exposure was determined using F-statistics. For the first released dataset and T2D summary statistics, the F-statistic was estimated using $\beta^2 / se^2$, where beta is the effect estimate and se its corresponding standard error. For the second release dataset, individual-level data from the UKBB and the R package ivreg (version 0.6–1) were used to calculate the F-statistic. Only IVs with an F-statistic larger than 10 were retained, which corresponds to a total of 5 to 75 IVs when using the mQTL statistics from Borges et al.[31] based on the first metabolite release, and 4 to 90 IVs when using the results from our mQTL analysis performed on the second metabolite release.

Data were harmonized using the function harmonise_data() with default parameters from the R package TwoSampleMR (version 0.5–6)[35], which was further used to run the MR analyses. RadialMR [36] (version 1.0) was used to assess heterogeneity and remove outliers for metabolites with a significant SNP heterogeneity. MR significance in the first release dataset was assessed based on the inverse variance weighted (IVW) method using q-values at a 5% threshold. Six additional MR methods (MR Egger, weighted median, simple mode, weighted mode, IVW fixed effect, IVW random effect and Steiger filtered IVW) were used as sensitivity analyses to check for concordant direction of causal effect estimate. These sensitivity analyses enable to evaluate causation under different MR assumptions, such as MR-Egger being robust to pleiotropy, or Steiger-filtered IVW removing IVs more strongly associated with the outcome than with the exposure to limit reverse causation effects. We only considered metabolites having a non-significant heterogeneity and pleiotropy estimates, measured by the Q-statistic and the MR-Egger intercept, respectively (5% significance threshold). Significant metabolites

were considered as replicated if they had an IVW q-value lower than 5% in the second release dataset with a concordant direction of effect of the IVW method with the first release dataset.

Statin medication, prescribed to lower LDL-related levels, is known to have an impact on T2D [37,38] and could therefore bias the MR results. To limit the impact of statin medication in our MR analyses, we therefore performed one-sample MR in the forward direction restricted to UKBB participants not taking any statin medication. For this, we removed individuals taking medication with an ATC code starting with C10A or C10B corresponding to 'lipid lowering agents'. We performed this sensitivity analysis in the second release set for the significant metabolites in the forward direction using the ivreg package (version 0.6–1) (96,380 individuals with 1,044 T2D cases and 95,336 controls). We used the same IVs as in the main MR analyses in the second release set and recomputed the F-statistics on the set of individuals not under statin medication to select only IVs with F>10. As for the other sensitivity analyses, we then compared the directions of effect.

**Interaction QTL.** We performed an interaction QTL analysis to investigate the genetic regulation of metabolite levels in T2D cases and controls using the REGENIE software with the--interaction option (version 2.2.4) [32] and the following interaction test:

$$y \sim SNP + T2D + Covariates + SNP*T2D$$

Where $SNP*T2D$ represents the interaction term between prevalent T2D disease status and genotypes. Only the p-value of the interaction test ("ADD-INT_SNPxVAR" column from the REGENIE output) was assessed for statistical significance. Interaction analyses were run on each of the two release datasets separately and were then meta-analyzed using the software METAL [39] to maximize statistical power. We considered suggestive variants if they had a meta-analysis p-value lower than the genome-wide significance threshold of $5x10^{-8}$, a nominally significant p-value in both release datasets, and a concordant direction of effect between both datasets. To investigate whether the results from the interaction QTL analyses could be due to the effect of confounding variables, we performed sensitivity analyses where we sequentially adjusted for BMI, lipid-lowering medication, and metformin (S1 Appendix). Lipid medication was considered as ATC code starting with C10A or C10B, and a binary variable was considered indicating if participants were taking any of this medication.

*Replication of interaction QTL effects in Estonian Biobank.* We replicated the interaction QTL analysis in the Estonian Biobank (EstBB), in which the same metabolite panel has been assayed, for the significant variants identified in the UKBB. The EstBB data freeze including altogether 211,728 biobank participants was applied.

The T2D cases were defined using the same approach as in the UKBB, where T2D diagnosis was based on the ICD10 codes E11* from electronic health registries (EHRs), and further on HbA1c levels for incident T2D cases. Individuals who have consumed insulin at least one year after the T2D diagnosis were excluded (EHRs do not have information about prescription of Insulins and analogies (the Anatomical Therapeutic Chemical code A10A*)). The control group was defined as follows: (1) they do not have ICD10 codes E10*, E11* or O24* marked in their EHRs, (2) they have not been prescribed any of the drugs with the following ATC codes: A10A*, A10BA02, A10BF01, A10BB07, A10BB03, V04CA01, A10BB01, A10BB09, A10BB12, A10BG01, A10BX02, A10BG03, A10BX03, A10BG02.

Similar to UKBB, metabolite data is available from Nightingale NMR spectrometry platform, and a QC was performed on both these data and the genotyping data (S1 Appendix). After QC, 6,237 T2D cases and 92,381 controls were enrolled into the replication analysis. Replication interaction QTL analysis was conducted by fitting a linear model with the interaction term between SNP and T2D status, while using SNP dosage, T2D status, sex, age at agreement,

spectrometer serial number and ten first genetic principal components as covariates on each pair of variant and metabolite. Analysis and data processing was implemented into custom scripts and by using R v4.3.1. We considered replicated signals those with a q-value lower than 5% in the EstBB cohort and with a concordant direction of effect.

**Analysis of T2D complications.** We analyzed the differences between metabolite levels and the four complication groups defined previously (microvascular, macrovascular, micro and macrovascular, no complications) using a multinomial approach. We used the R package mlogit (version 1.1–1), with T2D individuals without complications representing the reference level. To increase statistical power, we meta-analyzed the results across the two release datasets for each complication group. We declared metabolites as having a significant effect between different complication groups if they had a q-value lower than 5% in the meta-analysis, and a nominal significant p-value in each of the release datasets with a concordant direction of effect. Finally, we performed forward one-sample MR analyses within the UKBB to assess the causal effect of metabolite levels on the risk of developing T2D complications using the R package ivreg (version 0.6–1). We used the same metabolite IVs as for the T2D bi-directional two-sample MR, and only kept the ones having an F-statistic greater than 10 in the subset of T2D individuals.

## Results

### Causal associations

The upset plot in Fig 1 represents the causal relationships between T2D and metabolites from the bi-directional MR that were significant in the first release dataset and replicated in the second release dataset, and which passed all our sensitivity criteria. This includes no significant pleiotropy and no significant heterogeneity (assessed at 5%), and concordant directions of effect across six sensitivity methods (MR Egger, weighted median, simple mode, weighted mode, IVW fixed effect, IVW random effect and Steiger filtered IVW). Fig 1 also includes the significant associations between metabolite levels and prevalent/incident T2D in the UKBB

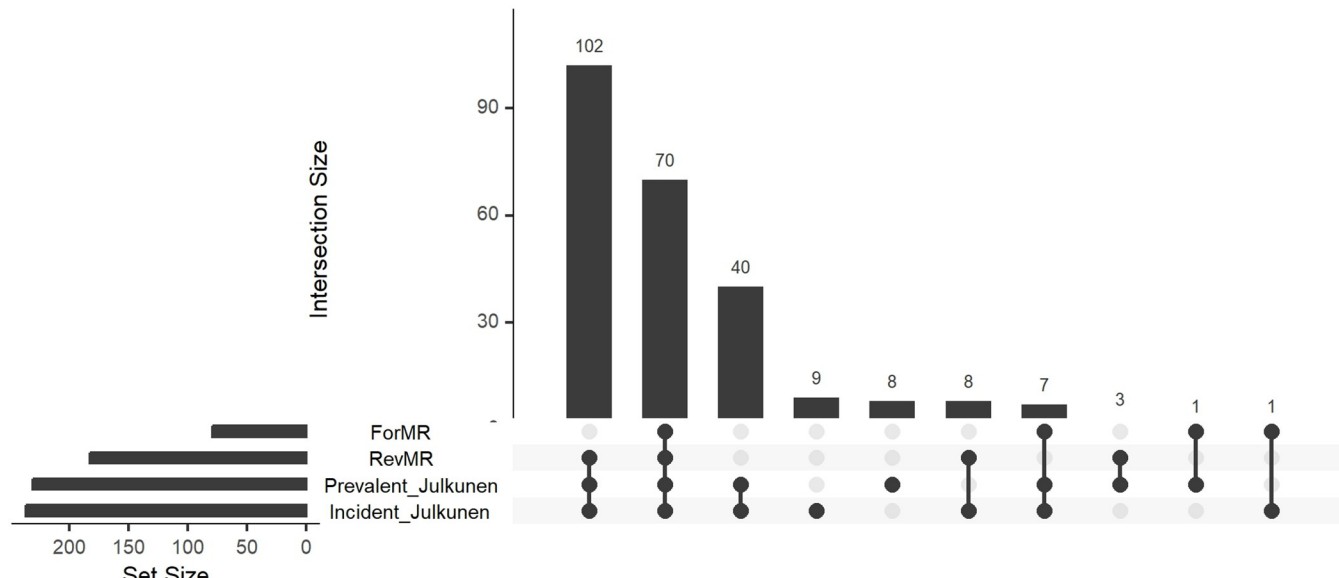

**Fig 1. Upset plot of the forward (ForMR) and reverse (RevMR) MR analyses, along with the association results from Julkunen et al. [4] between metabolite levels and prevalent (Prevalent_Julkunen) and incident (Incident_Julkunen) T2D.**

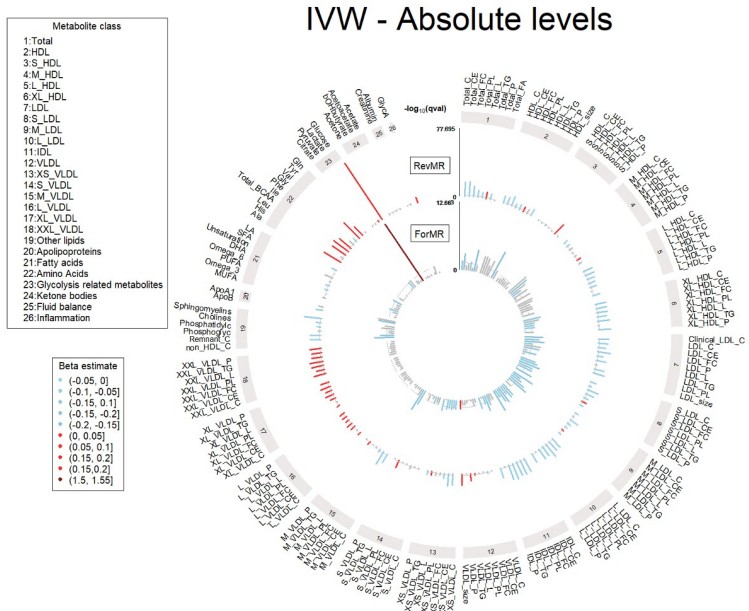

**Fig 2. Circular plot of the forward (ForMR) and reverse (RevMR) MR analyses with inverse variance weighted (IVW) estimates from the 1st metabolite set (absolute metabolite values).** Metabolites having increased levels associated with T2D occurrence are presented in red and the ones with decreased levels in blue. Non-significant metabolites, or metabolites not replicated in the 2nd metabolite set are colored in grey. Metabolites are grouped according to metabolite classes. The y-axis corresponds to the q-value (qval).

described by Julkunen et al.[4] using the first metabolite release dataset. All metabolites that showed a significant causal association in either MR direction in our analysis were also associated with prevalent or incident T2D reported by Julkunen et al.[4], supporting the observed MR effects. Interestingly, while among the metabolites significant in the reverse direction, i.e., causally affected by T2D liability, 172 were found to be associated with both incident and prevalent T2D, 8 were found to be associated only with incident T2D and not significant in the forward direction. This finding points to T2D predisposition affecting these 8 metabolites levels, which are all related to very low-density lipoproteins (VLDL). MR estimates from the IVW method in both directions are reported in Table C in S3 Appendix and shown in Figs 2 and 3 as circular plots for 168 absolute metabolite levels and 81 derived ratios.

We identified 78 metabolites significant in the forward direction, i.e., that showed a causal association with T2D risk, mostly spanning lipid classes, especially low-density lipoproteins (LDL) with decreased levels increasing the risk of T2D. While counterintuitive, these results are concordant with previous work using data from the UKBB [4,18], and a recent study describing genetic variants associated with both higher T2D risk and lower LDL levels [40]. Increasing evidence from observational studies also suggests that individuals with low LDL-C have a higher risk of developing T2D [41,42], these observations being retained when restricting the analyses to individuals not being under statin medication. Statin medication, used to lower LDL-C levels, is known to be associated with an increased risk of developing T2D [37,38]. To avoid potential bias in the MR analyses, we performed an additional sensitivity analysis in the forward direction in the second released metabolite restricted to individuals not taking statin medication through one-sample MR within the UKBB. We observed the same

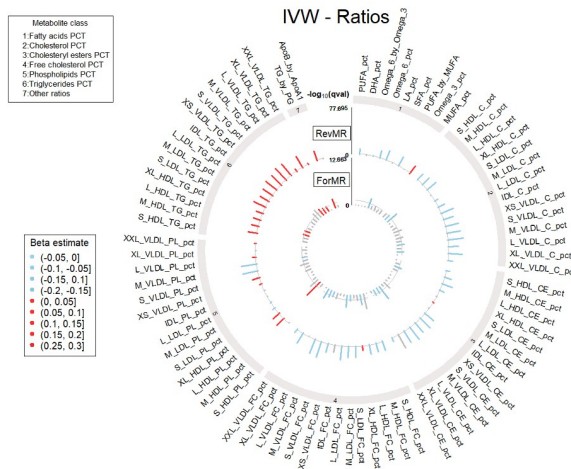

**Fig 3. Circular plot of the forward (ForMR) and reverse (RevMR) MR analyses with inverse variance weighted (IVW) estimates from the 1st metabolite set (metabolite ratios).** Metabolites having increased levels associated with T2D occurrence are presented in red and the ones with decreased levels in blue. Non-significant metabolites, or metabolites not replicated in the 2nd metabolite set are colored in grey. Metabolites are grouped according to metabolite classes. The y-axis corresponds to the q-value (qval).

negative direction of effect for all LDL-related metabolites, including LDL-C, which retained FDR significance (Table D in S3 Appendix). The magnitude of the effect was lower, which could be due to the lower statistical power of one-sample MR compared to two-sample MR using large T2D summary statistics. These results suggest that the causal effect of LDL-related metabolites is independent of statin medication, in line with previous studies showing that, while statins have a causal effect on LDL-C levels, there is no evidence for a direct causal impact on T2D risk [43,44]. Glucose was the strongest signal in the first dataset (OR = 4.57 [3.15–6.63], p-value = $8.73 \times 10^{-16}$), which was replicated in the second released dataset. Additionally, some ratios in high density lipoproteins (HDL) and VLDL classes were found to be causally associated with T2D risk. We found no evidence of a causal role of absolute triglycerides (TG) levels on T2D, although it has been described as a predictor of T2D risk [17,45]. We replicated findings from Mosley et al. [16], which did not find evidence of a causal role of BCAAs on the risk of T2D. Almost all of the significant metabolites in the forward direction were also found to be associated in the reverse direction, highlighting the complex interplay between metabolite profiles and T2D liability.

We found 183 metabolites to be significantly associated in the reverse direction, of which 114 were not found in the forward direction. This includes a negative effect of T2D liability on large and very large HDL, intermediate-density lipoproteins (IDL) and LDL, while the opposite trend is observed for large and extremely large VLDL. Consistent trends were observed for lipoprotein ratios with T2D liability being causally associated with decreases in cholesterol and cholesteryl esters fractions, and with increases in TG ratios. Fatty acids percentages were also affected by T2D liability, with decreases in PUFA and omega-6 percentages, but increases in MUFA percentage. Finally, we observed causal effects of T2D liability on all BCAAs, as well as on tyrosine and alanine. We further compared our results with a reverse MR study that was carried out by Smith et al. on the first release dataset of metabolites from the UKBB [18] using T2D summary statistics that included UKBB samples, and observed a high correlation of the effect estimates and p-values with our study (Fig B in S2 Appendix). Here, we report

replication of 93% of the findings from Smith et al. [18] in the second release dataset of metabolites from the UKBB without a potential bias due to sample overlap (Table C in S3 Appendix).

## Interaction QTL

We investigated whether there was a different effect of the genetic variants on metabolite profiles, which could explain the causal effects of T2D liability on metabolite profiles, through an interaction mQTL analysis. We identified 222 variants having a significant interaction between the mQTL effect and T2D status, corresponding to 14 metabolites, including glycine and various lipids. Of these, 40 pairs of variants were replicated in the EstBB (Table E in S3 Appendix), covering four metabolites: percentage of free cholesterol in small HDL (S_HDL_FC_pct), percentage of phospholipids in large LDL (L_LDL_PL_pct), percentage of triglycerides in large VLDL (L_VLDL_TG_pct) and percentage of free cholesterol in large VLDL (L_VLDL_FC_pct). These significant interactions mapped to two intergenic regions, one shared by all metabolites except S_HDL_FC_pct, and the second showing significant genetic interactions with T2D status only for S_HDL_FC_pct (Fig C in S2 Appendix). The genetic associations of these variants with metabolite levels are stronger in the control group than in the T2D patients' group, with some of the variants in the interaction regions being a significant mQTL only in the control group. For instance, rs6073958 is associated with lower levels of S_HDL_FC_pct in the controls group (beta = -0.146, p<1e-300), but not in the patients' group (beta = -0.002, p = 0.93) (Fig 4).

L_VLDL_TG_pct was significant in both MR directions, while S_HDL_FC_pct and L_VLDL_FC_pct were only significant in the forward and reverse MR analysis respectively (Fig 5). All of the interaction variants replicated in the EstBB are mQTLs for the corresponding metabolites in the overall UKBB cohort but are not associated with T2D risk in the latest T2D GWAS meta-analysis [3]. These findings suggest that the different genetic regulation of metabolite levels observed between T2D patients and controls is a consequence rather than a cause of the disease, in line with the results of the reverse MR for S_HDL_FC_pct and L_VLDL_TG_pct. This also goes in line with the observed interaction effects with the different genetic regulation of metabolites decreasing the differences in metabolite levels between cases and controls. The results remained consistent after adjustment for BMI, lipid-lowering medication or metformin medication (Table F in S3 Appendix).

To gain more insights into the link between the four replicated metabolites and T2D, we looked at the most significant interaction QTLs. The most significant interaction QTL for L_VLDL_TG_pct, rs3816117, is annotated by the variant effect predictor (VEP [46]) as a modifier variant for cholesteryl ester transfer protein, (CETP, Fig D in S2 Appendix), a protein targeted by drugs to increase HDL-C and decrease LDL-C levels. The significant interaction variant for S_HDL_FC_pct, rs6073958, is a QTL variant of the *PLTP* gene, which is involved in the transfer of phospholipids from triglycerides to HDL and might be involved in cholesterol metabolism.

## Metabolite profiles and T2D complications

We sought to better understand whether alterations in metabolite levels causally affected by T2D liability were associated with the risk of developing T2D complications (comparing T2D patients without complications, with microvascular complications, with macrovascular complications, and with both types of complications). Out of the 249 tested metabolites, 165 (66%) were associated with at least one of the complication groups, of which 156 were associated with the macrovascular group (Fig 6A). Only 3 and 6 signals were exclusive to the

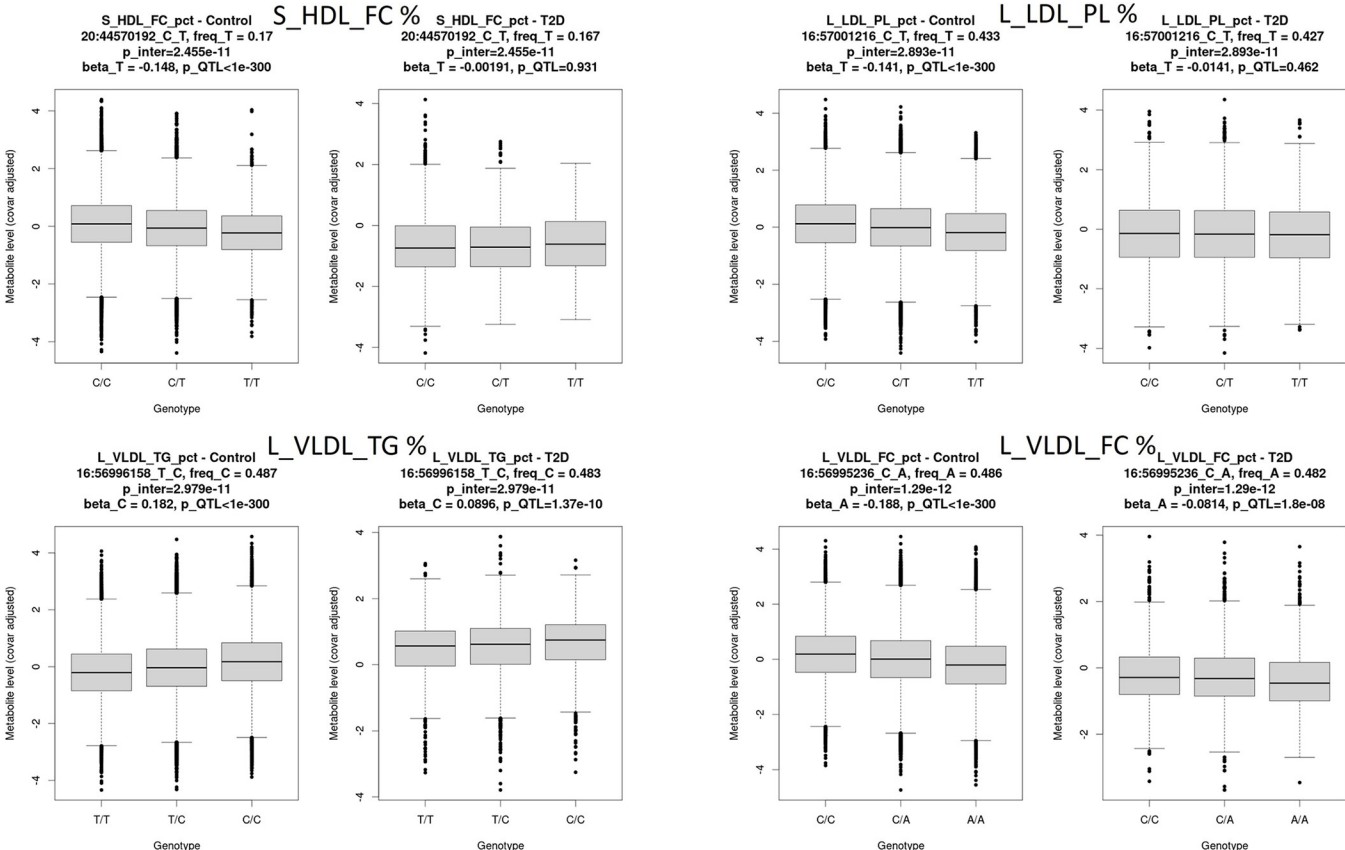

**Fig 4. Boxplot showing metabolite levels according to the genotype in T2D patients and controls (the most significant variant for each of the four metabolites replicated in the EstBB are represented).** The metabolite levels represented are inverse-normal transformed and adjusted for the covariates used in the interaction QTL analyses. For each tested variant (ID is in the form chr_pos_NonEffectAllele_EffectAllele), its frequency in the control and in the T2D patients' group is provided, along with the beta and p-value of association with metabolite levels. The overall p-value of the interaction test is also given.

'microvascular' and 'both complications' group, respectively. This pattern likely reflects the power of the analysis as the macrovascular group is three to five times larger than the two other complications groups (Table 1). Most of the significant metabolites were shared across multiple complication groups, showing that there is a metabolic signature associated with T2D complications.

Four metabolites were more strongly associated with microvascular complications than with macrovascular complications (Fig 6B). This is for example the case for creatinine, which was one of the strongest signals associated with microvascular complications (OR = 1.85 [1.58–2.18], p = $3.59 \times 10^{-14}$). Creatinine is used in estimated glomerular filtration rate (eGFR) calculation, a measure used to assess kidney function, and known to be affected in nephropathy which is one of the main T2D microvascular complications [47]. Leucine is another example, for which significant association was found for the microvascular group (OR = 0.76 [0.65–0.89], p = $7.33 \times 10^{-4}$), concordant with previous studies describing negative associations between leucine and kidney disease in T2D patients [48]. Increased levels of leucine were found to be putatively caused by an increased T2D liability in our reverse MR analysis, in line with the existing literature [6], showing opposite associations of leucine with T2D and its complications. Overall, we found the profile of associations with metabolites to be similar with the MR estimates observed with T2D liability, with 78% of the metabolites associated with at least

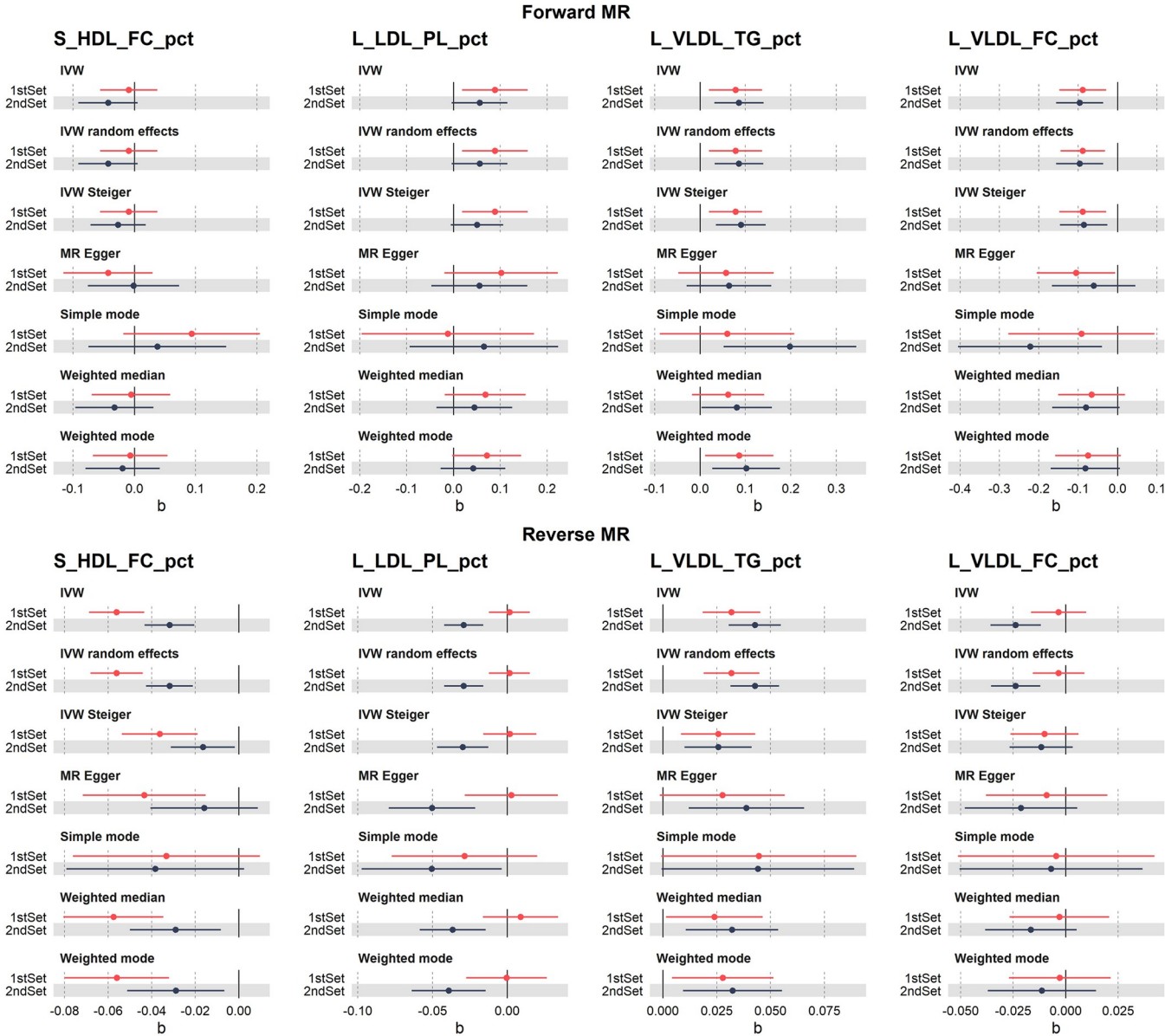

**Fig 5. Bi-directional MR results for the four metabolites significant in the interaction QTL analysis.** The top row corresponds to the forward MR (effects of metabolites on T2D), and the bottom row to the reverse MR (effects of T2D liability on metabolites). Effect sizes and standard error using the inverse variance weighted method (IVW) and the sensitivity methods are represented in red for the first set of metabolites, and in black for the second set of metabolites.

one complication and significant in the reverse MR having concordant direction of association between the two analyses, including decreases in most cholesterol metabolites, apolipoprotein B and glycine. Out of the 165 significantly associated metabolites, 54 and 118 were significant in the forward and reverse MR analyses, respectively. Of these, glycine and glycoprotein A were significant only in the reverse direction and associated with microvascular complications, suggesting that increased risk of these complications in T2D patients could be partly explained by an impact of the T2D liability on metabolite levels. A total of 40 metabolites were associated with complications but not significant in the MR analyses, including creatinine, glutamine, and apolipoprotein A1, which were all previously associated with microvascular complications

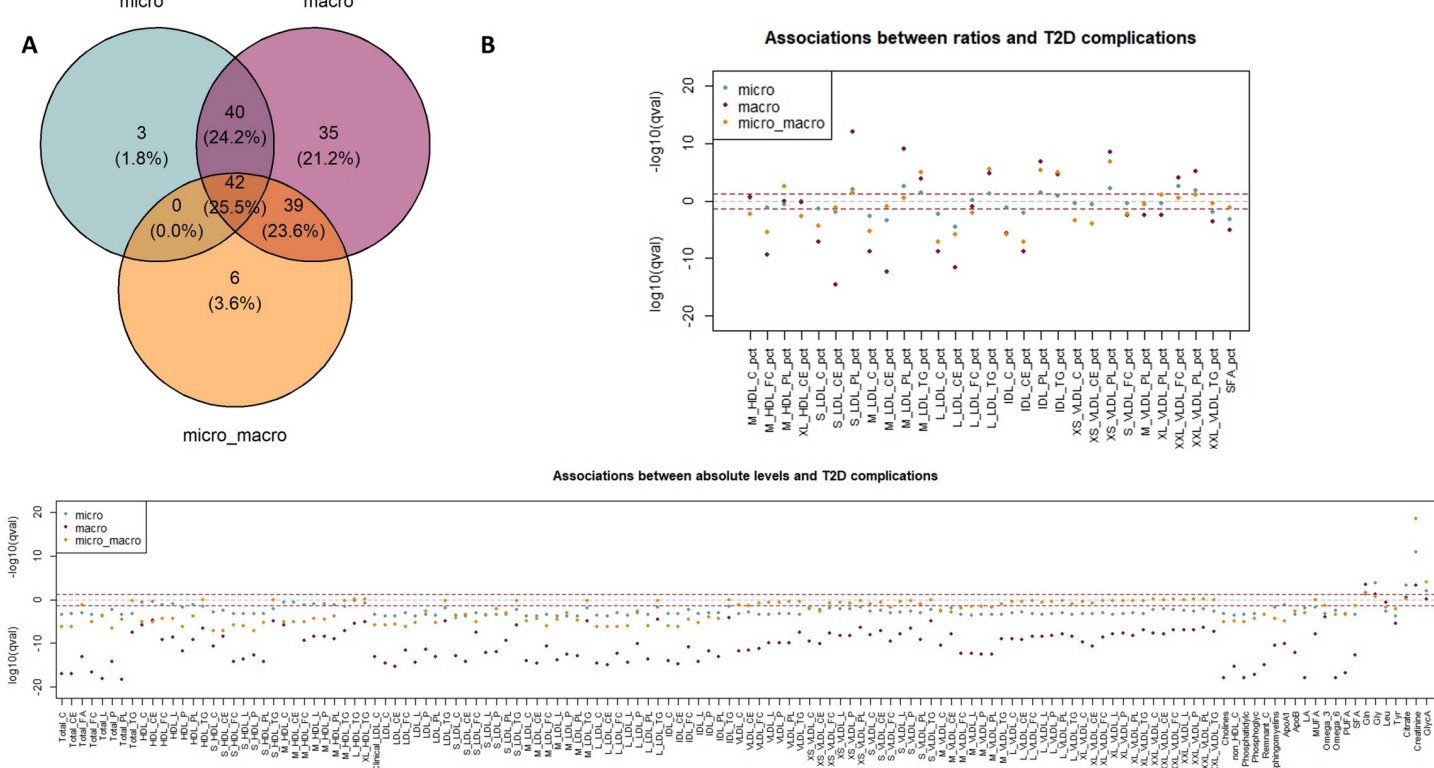

**Fig 6. Association analyses between metabolite levels and T2D complications (microvascular 'micro', macrovascular 'macro', or both complication groups 'micro_macro').** A: Venn diagram of the significant metabolites (meta-analysis q-values lower than 5%) across complications. B: log10(q-values) of associations between metabolite levels (separated into absolute levels and derived ratios) and each complication group.–log10(q-values) are represented for positive associations, and log10(q-values) are represented for negative associations.

in the literature [47,49,50]. This finding points towards molecular mechanisms involving these metabolites being more specific to disease complications rather than T2D itself.

The metabolite associations with T2D complications stayed stable upon adjustment, with 164, 158 and 156 signals remaining significant upon lipid medication, BMI and metformin adjustment, respectively (Table G in S3 Appendix). Metabolites affected by metformin and BMI adjustment span VLDL and HDL classes, while for instance valine, total VLDL size, albumin and total BCAAs became significant after adjustment. When applying one-sample MR between metabolite levels and the risk of T2D complications, we found no evidence of causal associations, potentially due to the low statistical power associated of the within-T2D UKBB MR analysis. Altogether, these results show that some metabolites causally affected by T2D liability are also associated with T2D complications, while others are specific to the development of complications, even though causality is still to be demonstrated.

## Discussion

In this study, we have investigated the links between plasma metabolite profiles, genetics, and the risk of T2D and subsequent complications, to evaluate the metabolic consequences of T2D. We have identified more metabolites as causally affected by T2D liability rather than having a causal effect on disease risk. Further, we have shown that T2D occurrence seems to have an impact on the genetic regulation of metabolite levels. Finally, we describe that metabolite

profiles are also associated with T2D complications, though no significant causal relationship could be demonstrated in the present study.

Among the 183 metabolites causally affected by T2D liability, we found positive estimates with BCAAs, as well as with alanine and tyrosine, in agreement with previous studies [13, 18, 26], but we did not find causal effects of BCAAs on T2D risk. Tyrosine was significant in both MR directions, but with opposite direction of effect, with decreased tyrosine levels being causally linked to increased T2D risk and T2D liability having a causal effect on increased tyrosine levels. These findings are in concordance with previous MR studies on the same cohort [26] but not with observational studies, which describe positive associations between tyrosine levels and T2D risk [6]. Furthermore, we found negative associations between tyrosine levels and the three complications groups tested, in line with a previous study [24], suggesting complex relationships between this metabolite and the risk of T2D complications. Further work is required to better disentangle the effect of tyrosine on T2D etiology. Glycine was significant only in the reverse MR analysis and associated with microvascular complications only. This amino acid has been already described as exhibiting lower levels in T2D patients, which could play a role in aggravating glucose dysregulation [51]. Even though not replicated in the EstBB, glycine presented the strongest signal in the interaction QTL analysis in the UKBB, with a weaker genetic regulation in T2D patients compared to controls. A genetic dysregulation of glycine levels might, therefore, be involved in T2D complications etiology, but replication is needed in an independent cohort.

Our findings also provide insights into the role of lipoprotein classes in T2D etiology. For example, TG levels have been described to be positively associated with T2D[40] and are known to positively correlate with glucose levels, a relationship that may be exacerbated in individuals with high polygenic risk scores for T2D[52]. TG levels, which we found to be causally affected by T2D liability and for some of the related percentages significant in our interaction QTL analysis, may therefore contribute to increased glucose levels. For all other types of lipoproteins that were significant in the reverse MR analyses, T2D liability had a causal effect on decreasing their levels. This includes absolute LDL levels and to a lesser extent their related percentages, in line with previous studies [4, 18, 41, 42, 53]. However, these relationships are still debated, and caution is to be taken when interpreting results in studies not restricted to medication-free individuals. For example, Smith et al.[18] showed in a similar reverse MR setting, using age as a proxy for medication use, that lipoprotein associations could be distorted upon medication use. Lipid-related medication adjusted T2D summary statistics are not available, and even if they were, care should be taken in interpreting results from such MR analyses using adjusted summary statistics due to potential collider bias [54]. It is to note that in our one-sample forward MR sensitivity analysis restricted to UKBB participants not under statin medication, LDL-related metabolites retained significance with the same direction of effect as in the two-sample MR analyses. Overall, we observed a causal effect of T2D liability on all lipid classes, including decreases in cholesterol levels, both in HDL and LDL, and increases in triglyceride levels, characteristic of the dyslipidemia observed in T2D[55].

We identified 165 associations between metabolite levels and the risk of developing T2D complications, most of which were shared across complication groups with only few, such as creatinine and glycine, being more strongly associated with microvascular than macrovascular outcomes despite smaller sample size of the microvascular group. A total of 118 metabolites were also found to be significant in the MR analyses, with similar profiles of association. A counter example of this is leucine, which showed associations in opposite directions between the reverse MR analysis and the differential level analysis with T2D complications. The discrepancies observed in our study between the two MR directions, as well as with the risk of complications, highlight the need for further work to better understand disease trajectories of T2D and its complications. A total of 40 metabolites were associated with at least one

complication group while not being significant in any of the MR analyses with T2D, suggesting mechanisms specific to T2D complications. However, causal inference to disentangle causation from association in the risk of T2D complications is warranted but challenging given the limited statistical power of MR analysis restricted to T2D patients.

In addition to investigating relationships between metabolite levels and T2D, we described for the first time, to our knowledge, differences in their genetic regulation between T2D patients and controls for 14 metabolites, of which four were replicated in an independent cohort. All of the significant variants identified were found to be significant mQTLs in the overall UKBB but were not associated with T2D risk [3]. These results suggest that deregulated levels of these metabolites are a consequence rather than a cause of the disease, in line with the observations from the bi-directional MR, where evidence is found in the reverse direction for two of these four metabolites. By performing an agnostic genome-wide scan to look for interaction signals, we have observed that all the interaction variants, which are mQTLs at a population-level, led to a decreased magnitude of metabolite genetic regulation in the T2D patients' group compared to the control group. Additionally, we observed that these interactions resulted in attenuated differences in metabolite levels between cases and controls. The decreased magnitude of genetic regulation in cases might be explained by feedback mechanisms trying to restore homeostasis equilibrium following the occurrence of the disease. Further research is needed to validate this hypothesis and to further interpret these findings, which is challenged by the fact that significant metabolites correspond to percentages and not absolute values. On the other hand, our findings could be of clinical relevance, as for example the interaction variants identified for the percentages of L_LDL_PL, L_VLDL_TG and L_VLDL_FC in our study are regulators of *CETP*, a target of lipid-lowering drugs, which have been shown to correlate with diabetes incidence [56]. On the other hand, variants in the interaction region for S_HDL_FC_pct are QTL variants of the *PLTP* gene, which is involved in the transfer of phospholipids from triglycerides to HDL and might be involved in cholesterol metabolism. Investigating the differential genetic regulation of metabolite profiles in T2D patients might therefore help in better understanding how lipid profiles are affected upon T2D, a question still debated across studies [55].

The present work presents some limitations. We have performed bi-directional MR using a similar framework to previous studies in the UKBB based on the first release dataset of metabolite data [16, 18], with the benefit of providing replication in the second release dataset. The use of the same outcome data for the two release datasets prevented the meta-analysis of the results. In this study, we were restricted to the European ancestry DIAMANTE study from 2018 because it was the latest one with summary statistics available without the UKBB samples, which enables us to avoid sample overlap between the exposure and outcome and to limit potential related bias [12]. Our results need to be extended to non-European populations, which will require global efforts to characterize the genetic regulation of molecular traits in these populations, along with methodological developments to deal with multi-ancestry data, especially in MR studies. Additionally, our findings warrant replication in cohorts external to the UKBB, especially for the MR analyses and complication associations. Finally, our study provides useful insights into the metabolic consequences of T2D but is limited by the assayed metabolite panel, which is mostly composed of lipid-related plasma metabolites. Additional metabolite classes, as well as metabolite levels from different tissue types would help in better unravelling biological mechanisms. Furthermore, metabolites were considered individually as exposure, which does not capture overall metabolic pathways. As expected, we observed similar causal effects of highly correlated metabolites belonging to the same class. It would be of interest to integrate metabolites together, for example through multivariate MR analyses [57] within each metabolite class, to assess their relative contribution in shaping T2D risk. In this

study, we have here addressed the potential confounding effect of lipid-lowering medications in our MR analyses using restricted one-sample MR. It would however be interesting to use multivariate models to include variables such as BMI to better understand the biological pathways mediating causal relationships between T2D and metabolite profiles.

While MR studies enable the assessment of causality between an exposure and an outcome, they are prone to false positives and based on the liability of the disease but not its occurrence. By using interaction models, we went one step further to describe the consequences of T2D occurrence on metabolite levels and their genetic regulation. We highlight that changes in metabolite profiles can be useful to better understand T2D disease progression, as exemplified by the metabolites associated with the risk of developing T2D complications. Altogether, our results enable a deeper understanding of the metabolic consequences of T2D and provide future directions for the study of the genetic regulation of molecular levels in T2D and its complications to better capture disease trajectory.

## Supporting information

**S1 Appendix. File containing additional details on the quality control and sensitivity analyses performed.**
(DOCX)

**S2 Appendix. Containing Supplementary Figures A to D.** Fig A: Workflow to define prevalent T2D cases at T0, when metabolite data have been measured. Fig B: Comparison of betas and p-values with Smith et al. 2023 for the reverse MR on the first set of metabolites [18]. Fig C: Manhattan plot of the interaction QTL analysis for the four metabolites significant and replicated in the EstBB. Fig D: Genomic regions around the most significant variants from the interaction QTL analyses, obtained from https://genome.ucsc.edu/.
(DOCX)

**S3 Appendix. Containing the Supplementary Tables A to G.** Table A: list of ICD10 codes used to define the T2D complication groups. Table B: summary of the analyses performed with the number of individuals kept, the covariates used for the adjustment and the significance threshold. Table C: Results of the bi-directionnal MR analyses (IVW method) in the 1st and 2nd set of metabolites. Table D: Results of the one-sample MR Forward analysis (ivreg estimate) in the 2nd release set of significant metabolites from the main MR analyses. Table E: Variants significant at the genome-wide significance threshold (5e-8) in the interaction analysis and replicated in the Estonian Biobank (q-value <0.05). Table F: Sensitivity analyses for the replicated variants from the interaction QTL analysis. Table G: Results of associations between metabolite levels and the three complication groups (microvascular, macrovascular, and both micro and macro vascular).
(XLSX)

## Acknowledgments

This research has been conducted using the UK Biobank Resource under Application Number 10205.

The Estonian Biobank research team refers to Andres Metspalu, Lili Milani, Tõnu Esko, Reedik Mägi, Mari Nelis and Georgi Hudjashov.

Data analysis in the EstBB was carried out in part in the High-Performance Computing Center of University of Tartu. The activities of the EstBB are regulated by the Human Genes Research Act, which was adopted in 2000 specifically for the operations of the EstBB.

## Author Contributions

**Conceptualization:** Ozvan Bocher, Ana Arruda, Eleftheria Zeggini.

**Data curation:** Ozvan Bocher, Archit Singh, Yue Huang, Ene Reimann, Andrei Barysenska, Anastassia Kolde.

**Formal analysis:** Ozvan Bocher, Urmo Võsa.

**Funding acquisition:** Reedik Mägi, Eleftheria Zeggini.

**Methodology:** Ozvan Bocher.

**Resources:** Urmo Võsa, Anastassia Kolde, Nigel W. Rayner, Tõnu Esko, Reedik Mägi.

**Supervision:** Eleftheria Zeggini.

**Visualization:** Ozvan Bocher.

**Writing – original draft:** Ozvan Bocher.

**Writing – review & editing:** Ozvan Bocher, Archit Singh, Yue Huang, Urmo Võsa, Ene Reimann, Ana Arruda, Andrei Barysenska, Anastassia Kolde, Nigel W. Rayner, Tõnu Esko, Reedik Mägi, Eleftheria Zeggini.

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
