## [Decision Letter · Decision Letter 0]

12 Aug 2024

Dear Dr Bocher,

Thank you very much for submitting your Research Article entitled 'Insights into the metabolic consequences of type 2 diabetes' to PLOS Genetics.

The manuscript was fully evaluated at the editorial level and by independent peer reviewers. The reviewers appreciated the attention to an important problem, but raised some substantial concerns about the current manuscript including additional details of the methods used, interpretation of analyses, interpretation of existing literature in the context of your findings, nomenclature, and novelty. Based on the reviews, we will not be able to accept this version of the manuscript, but we would be willing to review a much-revised version. We cannot, of course, promise publication at that time.

If you decide to revise the manuscript for further consideration at PLOS Genetics, please aim to resubmit within the next 60 days, unless it will take extra time to address the concerns of the reviewers, in which case we would appreciate an expected resubmission date by email to plosgenetics@plos.org.

To resubmit, log into your Editorial Manager account and select the option 'Revise Submission' in the 'Submissions Needing Revision' folder.

We are sorry that we cannot be more positive about your manuscript at this stage. Please do not hesitate to contact us if you have any concerns or questions.

Yours sincerely,

Themistocles L. Assimes

Academic Editor

PLOS Genetics

Scott Williams

Section Editor

PLOS Genetics

Reviewer's Responses to Questions

**Comments to the Authors:**

Reviewer #1: In the manuscript entitled “Insights into the Metabolic Consequences of Type 2 Diabetes,” the authors aim to link diabetes, metabolites, and the genetic component. This study primarily replicates previous findings rather than reporting new ones. The following points could be considered to improve the manuscript's quality:

1. In the abstract the authors are using metabolomics measurements from Nightingale platform which is limited in terms of the number of measured metabolites and is more lipoproteins oriented. Thus, statement such as: “Overall, this work provides a map of the metabolic consequences” might be misleading especially for people who are not familiar with the field of metabolomics.

2. In the abstract the authors are reporting on 165 metabolites identified to be associated with T2D complication. To bring the reader perspective, it would be important to mention out of how many.

3. In the abstract the authors are writing that mostly spanning lipid-related classes but the method is measuring mainly lipids and lipoproteins so it is not surprising that such observation was made. It would be suggested to differently address this.

4. In the introduction short references to the Nightingale platform briefly describing the number of molecule measured and nature of those molecules mentioning that mainly lipoproteins are measured would be beneficial for the reader. Including this information in the methods only is not sufficient.

5. For the methods. Although the Mendelian randomization is extensively described following points could be addressed:

A) Specify Software Versions: Include the specific versions of the software and libraries used, e.g., "We used the MendelianRandomization library (versionX) and the TwoSampleMR library (versionX)."

B) Although the authors describe that they are using Borges summery statistics. The clear specify the sources of the datasets and context on why these sources were chosen is missing.

C) Same for REGENIE software more clarity regarding this should be also added.

D) It would be recommended to describe instrument selection criteria in greater detail by including significance thresholds and LD clumping parameters.

E) The potential confounders could be better addressed by stating how potential confounders are handled or filtered out during the IV selection process.

F) The process of harmonizing exposure and outcome data before analysis should be better described.

G) More details on the sensitivity analyses performed to check the robustness of the results should be included.

H) It would be recommended to provide details on how heterogeneity and pleiotropy were assessed and mention the specific statistics used.

I) If applicable, explain how the analysis is performed when only a single genetic variant is available.

6. Given that the study was conducted in two big cohort, the potential clinical relevance of the discovery could be addressed in the discussion.

Reviewer #2: The authors conducted an extensive bi-directional Mendelian randomization (MR) study investigating the relationship between circulating metabolite levels and type 2 diabetes (T2D) using metabolomics data from the UK Biobank along with T2D summary statistics. Additionally, they performed an interaction QTL analysis within the UK Biobank on metabolite levels, using diabetes as an interaction variable, and examined complications.

Study Design and Results:

The study design of the MR is generally well-articulated, and the results are clearly presented. However, I have a few remarks.

Firstly, the authors mentioned using existing summary statistics from Borges et al., but the reference cited corresponds to a different study by this author. I assume the correct reference is:

- Borges MC, et al. Role of circulating polyunsaturated fatty acids on cardiovascular diseases risk: analysis using Mendelian randomization and fatty acid genetic association data from over 114,000 UK Biobank participants. BMC Med. 2022 Jun 13;20(1):210. PMID: 35692035.

Please confirm this reference.

Figure 1 indicates that all identified associations in the reverse and forward MR were previously reported by Julkunen. While the lack of novelty in the reported numbers is not a major issue, given that the authors aim to disentangle the causal direction of association (metabolite → diabetes and diabetes → metabolite), there are concerns regarding the interpretation of the data in a causal context:

1. Strong causal effect of low LDL on increased T2D risk.

2. Diabetes causally lowering many cholesterol-related metabolites.

The authors explain the first observation by referencing studies that suggest low LDL is associated with increased diabetes risk, but they fail to consider a possible bias introduced by statins, which are known to influence both metabolite levels (LDL) and the outcome (diabetes). Instead, the authors justify this observation using other studies:

- The first study shows an inverse association between LDL GRS and diabetes in individuals not on lipid-lowering medications. However, the 31 SNP LDL GRS included HMGCR, the target of statins, implying a genetically induced statin-like effect.

- The second study is an observational study with Vanderbilt EHR data, which acknowledges potential exposure misclassification due to missing statin exposure data across multiple healthcare providers, despite conducting a sensitivity analysis.

- The third study indicates that familial hypercholesterolemia is associated with lower diabetes risk, which is not directly comparable.

The lack of discussion on the potential impact of statin-induced diabetes is concerning.

Regarding the second observation, diabetes primarily lowering all types of cholesterol-related metabolites raises questions about the validity of the causal associations presented. Generally, diabetes leads to dyslipidemia rather than lowering cholesterol levels.

Interaction QTL Analysis:

The authors discuss effect mediation but present data on effect modification (interaction). The terminology should be corrected to accurately describe the analyses conducted. The interaction analysis lacks sufficient detail for proper interpretation, as only interaction coefficients are presented without the main effects. This omission hinders the interpretation of the interaction coefficients.

For example, ensuring that both independent variables (SNP and T2D) are oriented in the same direction concerning the risk of the investigated metabolite is crucial for inferring the interaction coefficient.

- Metabolite = SNP + T2D + SNP*T2D

It is assumed that disease status refers to prevalent T2D (not incident or any T2D). The interaction coefficient beta is interpretable only if both independents are oriented towards the same direction in predicting the outcome. Interaction effects can also be in the opposite direction, indicating that the combined effects of two risk factors are less than expected if isolated, often due to shared biology.

The authors claim to have identified 14 interaction QTLs, of which 40 were replicated in ESTBB. This discrepancy suggests that 40 variants were identified in the discovery analysis (as shown in the supplemental tables), with only 4 being replicated and shown in the box plots.

It is notable that all four replicated metabolites are percentages rather than absolute levels. For instance, the percentage of free cholesterol in small HDL shows higher levels in controls, with the T-allele increasing this metabolite, while in cases, the metabolite is lower and unaffected. Conversely, in L_LDLPL, LDL levels are higher in controls than in cases, and the interaction attenuates the T-allele effect, bringing it closer to the metabolite value in cases. Here, the interaction reduces the differences between groups rather than amplifying them.

The use of percentages complicates the interpretation further.

Minor Comments:

- In Figure 2, consider using red for color coding instead of yellow for better clarity.

Reviewer #3: This manuscript (MS) aims to elucidate the consequences of type 2 diabetes (T2D) by exploring the connections between metabolite profiles, genetic factors, and T2D risk and subsequent complications. The study investigates these relationships using data from two distinct biobanks: the UK Biobank and the Estonian Biobank. The topic is both timely and relevant, with the MS employing robust methodologies. Additionally, the MS is well-written and neat. However, it could benefit from further clarifications in certain areas.

Mayor

The manuscript title could be enhanced by incorporating more details about the methodologies employed. This would provide readers with a clearer understanding of the approach and scope of the study, making the title more informative and engaging.

The abstract could be improved by including more detailed information about the methodologies used to investigate the sources of the observed causal associations. Providing insights into the specific analytical approaches, data sources, and techniques employed will offer readers a clearer picture of how the study addresses its research questions and supports its findings. Additionally, the abstract mentions “lipid” classes but would benefit from specifying the particular lipid classes.

The manuscript focuses on a single metabolomic measurement as the exposure variable. This approach has important implications for causal inference. Relying on just one metabolite may limit the study's ability to fully capture the complexity of metabolic pathways associated with T2D risk. It might be beneficial to discuss the implications of this limitation.

The MS uses the term "mediation" in reference to genetic factors. However, based on the context provided, it appears that the authors are describing an interaction effect, which is more indicative of "effect modification" rather than traditional mediation analysis. Mediation typically involves exploring the mechanism through which one variable affects another through an intermediary variable, whereas effect modification refers to how the relationship between two variables changes across different levels of a third variable. It would be beneficial for the authors to clarify this distinction to ensure that the terminology accurately reflects the analytical approach used.

Please provide a rationale for retaining related samples in the analysis.

In line 94, the term “homogeneity” is used to partially justify the exclusion of non-European samples. Please avoid using this term in contexts where there are multiple sources of genetic and non-genetic heterogeneity, even among samples of European ancestry. It is crucial to recognize that variability can still exist within genetically similar samples due to other contributing factors across all populations.

Minor

The size of the figures makes them difficult to read. Increasing the resolution or size of the figures would improve their clarity and readability.

In the Results and Discussion sections, some verbs are written in the present tense. Please ensure that the tense is consistent with the rest of the manuscript, typically using past tense for describing the results and discussion of completed experiments.

Please include a description of all abbreviations used in the figures to enhance readability and ensure that the figures are easily interpretable.

**Have all data underlying the figures and results presented in the manuscript been provided?**

Reviewer #1: **No: **Summary statistics of the mQTL analysis in the second data release of the UKBB have been submitted by authors

to the GWAS catalog and will be released upon publication.

Reviewer #2: Yes

Reviewer #3: Yes

PLOS authors have the option to publish the peer review history of their article (what does this mean?). If published, this will include your full peer review and any attached files.

Reviewer #1: No

Reviewer #2: **Yes: **Marijana Vujkovic

Reviewer #3: No

---

## [Decision Letter · Decision Letter 1]

30 Oct 2024

Dear Dr Bocher,

We are pleased to inform you that your manuscript entitled "Disentangling the consequences of type 2 diabetes on targeted metabolite profiles using causal inference and interaction QTL analyses" has been editorially accepted for publication in PLOS Genetics. Congratulations!

Yours sincerely,

Themistocles L. Assimes

Academic Editor

PLOS Genetics

Scott Williams

Section Editor

PLOS Genetics

Aimée Dudley

Editor-in-Chief

PLOS Genetics

Anne Goriely

Editor-in-Chief

PLOS Genetics

Comments from the reviewers (if applicable):

Reviewer's Responses to Questions

**Comments to the Authors:**

Reviewer #1: The authors responded to all my comments and suggestions.

Reviewer #2: Thank you for addressing my comments, providing clarifications and substantiating your claims with additional sensitivity analyses, i have no openstanding remarks.

Reviewer #3: Authors have addressed my comments accurately.

**Have all data underlying the figures and results presented in the manuscript been provided?**

Reviewer #1: Yes

Reviewer #2: Yes

Reviewer #3: Yes

PLOS authors have the option to publish the peer review history of their article (what does this mean?). If published, this will include your full peer review and any attached files.

Reviewer #1: No

Reviewer #2: No

Reviewer #3: **Yes: **Magdalena Sevilla-Gonzalez

**Data Deposition**

http://datadryad.org/submit?journalID=pgenetics&manu=PGENETICS-D-24-00669R1

**Press Queries**

---

## [Editor Report · Acceptance letter]

26 Nov 2024

PGENETICS-D-24-00669R1 

Disentangling the consequences of type 2 diabetes on targeted metabolite profiles using causal inference and interaction QTL analyses 

Dear Dr Bocher, 

We are pleased to inform you that your manuscript entitled "Disentangling the consequences of type 2 diabetes on targeted metabolite profiles using causal inference and interaction QTL analyses" has been formally accepted for publication in PLOS Genetics! Your manuscript is now with our production department and you will be notified of the publication date in due course.

With kind regards,

Anita Estes

PLOS Genetics

On behalf of:
